# Obscurable Fishermen

## Abstract

In the process of applying for a job across several similar firms, applicants often have the option to exclude certain features from a CV, e.g., photo, GPA, standardized test scores, etc. If applicants desire the best income offer possible and can submit multiple applications to similar positions, they may exclude or include various of these optional features on different applications to see which yields the best results, eventually accepting the highest offer. But if an analyst then would like to estimate what makes a good worker using the applications (features) and incomes (outcomes) of the finally accepted offers, she will have an endogeneity problem! The excluded features, which we term "obscured" will be missing not at random, meaning simple imputation methods such as the conditional expectation will result in biased estimates. We formalize this problem and present a preliminary result in which we reduce our obscured setting to a high-dimensional instantiation of the setting from Cherapanamjeri et al. [1]. Unfortunately, this reduction increases the number of variables by an amount combinatorial in the dimension of the problem, meaning the algorithmic tool for this setting will not be efficient in the original parameters. We present possible next steps such as approximate SGD on the MLE and kernelization to get around the increase in variables.

## 1   Introduction

Borrowing the fisherman occupation from Roy [5], we introduce what we call our "obscurable fisherman" problem much like Cherapanamjeri et al. [1]:

Suppose agents in a small village have only *one* industry available to them: fishing. They may apply to be fishermen at various very similar firms and receive an offer of income according to some common policy based on the features presented in their application. However, all applications may *optionally* include FAT (Fishing Aptitude Test) scores among other features. Every agent sends applications to various firms including/excluding various optional features. Because all agents test the waters by including/excluding features, to calculate job offers, firms just take their best guess (a conditional expectation) as to what the values of the missing features are on each application. Eventually, each agent accepts the fisherman job offer the gives the highest income.

A statistician gets access to accepted offers and applications (with obscured values). She asks:
*What makes a good fisherman?*

This anecdote sets up an endogeneity problem for the statistician. If applicants desire the best income offer possible and can submit multiple applications to similar firms, the application of the offer they eventually select will have *strategically* obscured features. These features are missing not at random (MNAR)[6] and so should not be thrown out or imputed with conditional averages by the statistician. Are there algorithms efficient in time and sample complexity that the statistician can use? In Section 2, we formally present a model of this strategic feature obscuration, which we call the obscurable fisherman setting. The statistician must estimate $\mathbf{w} \in \mathbb{R}^d$, the coefficients of a linear policy assigning income offers based on features. Any subset of the first $k$ features can be obscured. Agents may test every possible obscuration pattern and then accept the one that yields the highest

(noisy) outcome. In Section 3, we present preliminary results as to the estimation of $\mathbf{w}$ using the linear estimation under model self-selection tools from Cherapanamjeri et al. [1]. Our obscurable fisherman setting, while a strategic missing data problem, can also be viewed as a model selection problem. As such, we create a reduction of a generic obscurable fisherman dataset, $D$, to a "good fisherman" (à la Cherapanamjeri et al. [1]) dataset, $\tilde{D}$, that is the best response to a set of models in the form of their setting. Unfortunately, the reduction requires an increase in the number of features that is combinatorial in the original dimension. Thus, directly using the algorithm they present is not efficient in the parameters of the obscurable fisherman setting. In Section 4 we discuss future work we hope will yield better results.

## 1.1 Related works

Cherapanamjeri et al. [1] is the most direct inspiration for our model; they consider agents who select (using a function such as max) between $k$ linear models and a statistician that estimates the $\mathbf{w}_j^\star$ coefficient for each model. In our version, there is only one underlying linear coefficient vector, $\mathbf{w}$, and instead agents select from obscuration patterns. The strategic selection of obscuration patterns means that we consider estimation under missing not at random data (MNAR) which was first formally defined by Rubin [6] and cannot generally be fixed with imputation of conditional averages. See Little [3] for a taxonomy and survey of estimation methods under various missing data patterns. Additionally, while we focus on a linear coefficient statistical estimation problem, there are similar questions that involve creating an optimal *classifier* given strategically obscured data. Krishnaswamy et al. [2] design classification algorithms that perform well under strategically obscured data and Liu and Garg [4] evaluate whether it is possible to build a classifier that does not implicitly penalize agents who choose to obscure test score data in university admissions.

## 2 Model

### 2.1 Agents

Each agent (she), $i \in [n]$ has feature vector: $\mathbf{x}^{(i)} \in \mathbb{R}^d$ drawn from a joint distribution $\mathcal{D}(\mathbf{x})$. The first $k < d$ of $d$ features are optional. That is, features at any subset $\mathcal{O}_j \subseteq [k]$ of indices may be obscured. We will call $\mathcal{O}_j$, a set obscured indices, an *obscuration pattern*. Let $\mathcal{O}_j \in \mathcal{P}$ where $\mathcal{P}$ is the set of all obscuration patterns. Clearly, $|\mathcal{P}| = \sum_{l=0}^{k} \binom{k}{l}$.

**Definition 2.1** (Obscured feature vector). *For a true feature vector, $\mathbf{x}^{(i)}$, and obscuration pattern, $\mathcal{O}_j$, an* obscured feature vector $\mathbf{x}_j^{(i)} \in \mathbb{R}^d$ *is the same as $\mathbf{x}^{(i)}$ except all elements at indices in the obscuration pattern are obscured. Formally: $x_{j,u}^{(i)} = x_u^{(i)} \ \forall u \in [d] \setminus \mathcal{O}_j$ and $x_{j,h}^{(i)} = o \ \forall h \in \mathcal{O}_j$. Where $o$ (for obscured) indicates that this a missing value and holds no inherent numerical meaning.*

When $o$s are replaced with conditional expectations:

**Definition 2.2** (Expected feature vector). *For an obscured feature vector, $\mathbf{x}_j^{(i)}$, and obscuration pattern, $\mathcal{O}_j$, an* expected feature vector, $\hat{\mathbf{x}}_j^{(i)} \in \mathbb{R}^d$, *is the same as $\mathbf{x}^{(i)}$ except all elements at indices in the obscuration pattern are expectations conditioned on all unobscured variables. Formally: $\hat{x}_{j,u}^{(i)} = x_u^{(i)} \ \forall u \in [d] \setminus \mathcal{O}_j$ and $\hat{x}_{j,h}^{(i)} = \mathbb{E}[x_h | U(\mathcal{O}_j)] \ \ \forall h \in \mathcal{O}_j$ where $U(\mathcal{O}_j)$ are the elements at unobscured indices, i.e., $U(\mathcal{O}_j) := \{x_u^{(i)} | u \in [d] \setminus \mathcal{O}_j\}$*

The agent privately tests a given linear model on each *expected feature vector* and selects the best outcome and obscuration pattern. That is she selects:

$$y^{(i)} := \max_{j \in |\mathcal{P}|} f_j(\mathbf{x}^{(i)}); j^{\star(i)} := \arg\max_{j \in |\mathcal{P}|} f_j(\mathbf{x}^{(i)}) \quad \text{where} \quad f_j(\mathbf{x}^{(i)}) := \mathbf{w}^\top \hat{\mathbf{x}}_j^{(i)} + \varepsilon_j$$

Noise $\varepsilon_j \sim \mathcal{N}(0, \sigma^2)$ is iid and drawn separately for each model. Notice that obscuration pattern and model are functionally the same. That is, if an agent chooses obscuration pattern $j$, she has chosen model $j$. We will use these terms interchangeably.

### 2.2 Learner

The learner (he) receives a dataset of the selected *obscured feature vectors* and best outcomes:

$$D := \{\mathbf{x}_{j^\star}^{(i)}, y^{(i)}, j^{\star(i)}\}_{i \in [n]}$$

85   First, note $D$ will have data that is missing not at random (MNAR). Second, note that the obscuration

86   pattern can be directly gleaned from $\mathbf{x}_{j^\star}^{(i)}$, thus receiving an obscured feature vector also allows the

87   learner to know which model was selected, $j^\star$.

88   The learner would like to know what makes a good outcome, i.e., estimate $\mathbf{w}$, despite the non-

89   randomness of the missing data. It is clear to see that the obscured setting creates endogeneity due to

90   correlated errors and thus standard OLS estimates (with either conditional expectation imputations or

91   dropping of missing data) would be biased.

92   **Example 2.1** (Learner does biased OLS)**.** *Suppose* $\mathbf{w} := (1,1)$, $\sigma^2 = \frac{1}{5}$, *and both* $x_1, x_2 \sim$

93   $\textsc{Unif}(-1,2)$. *Thus,* $x_1, x_2$ *are independent of one another and* $\mathbb{E}[x_2, |x_1 = x_1^{(i)}] = .5$ *for all* $x_1^{(i)}$.

94   *We simulate* $n = 200$ *of this example and imagine the learner does OLS on the full data set (i.e.*

95   *allowing* $o = .5$*) and also on just the points that have no missing data. This is presented in Figure 1.*

96   *Clearly both OLS estimators are biased.*

97   What time and sample efficient algorithms may the learner run such that he achieves an $\varepsilon$-unbiased

98   estimator of $\mathbf{w}$ despite strategically obscured data?

# 3   A reduction to Cherapanamjeri et al. [1] self-selection

100   In these results, we will detail a (relatively inefficient) approach to estimating $\mathbf{w}$ when conditional

101   expectations are known using existing model selection tools from Cherapanamjeri et al. [1]. Improved

102   methods and future work are discussed in Section 4.

103   **Assumption 3.1** (Known Conditional Expectations)**.** $\mathbb{E}[x_h|U(\mathcal{O}_j)]$ *is known* $\forall h \in \mathcal{O}_j, \forall \mathcal{O}_j \in \mathcal{P}$

104   Assumption 3.1 is a strong assumption stating that the expectation for all obscurable features condi-

105   tioned on any possible set of unobscured features is known.

## 3.1   Constructing a good fisherman setting

107   In the known-index model selection setting of Cherapanamjeri et al. [1], agents select a linear model,

108   $f_j(\mathbf{x}) = \mathbf{w}_j^{\star\top}\mathbf{x}^{(i)} + \varepsilon_j$, that provides the best sampled outcome. Importantly, the resulting dataset

109   provides $\{\mathbf{x}^{(i)}, y^{(i)}, j^{\star(i)}\}_{i \in [n]}$. Thus, while the provided *outcome* depends on the selected model,

110   the *feature set* does not. We will transform our learner's dataset, $D$, which contains the problematic

111   $\mathbf{x}_{j^\star}^{(i)}$ obscured features, into $\tilde{D}$, a dataset that could have come from a good fisherman setting. In

112   constructing $\tilde{D}$ we will shift each obscurable feature such that $w_h x_h \geq 0 \forall h \in [k]$. Thus we need:

113   **Assumption 3.2** (Obscurable features are sufficiently bounded)**.** *The following must hold for all*

114   *obscurable feature indices,* $h \in [k]$*: If* $w_h > 0$ *then* $l_h \leq x_h \quad \forall x_h$. *If* $w_h < 0$ *then* $u_h \geq x_h \quad \forall x_h$

115   **Definition 3.1** ($\tilde{D}$, good fisherman transformed dataset)**.** $\tilde{D} := \{1, \tilde{\mathbf{x}}^{(i)}, y^{(i)}, j^{\star(i)}\}_{i \in [n]}$ *where: each*

116   $\tilde{\mathbf{x}}^{(i)} \in \mathbb{R}^{g(k,d)}$, $g(k,d) := k\sum_{l=0}^{k-1}\binom{k-1}{l} + d$ *and is constructed according to Algorithm 1*

117   Notice that $\tilde{\mathbf{x}}^{(i)}$ no longer depends on the model selection! The constructed feature set is the original

118   with two key changes: (1) a shift on obscured variables (2) $k\sum_{l=0}^{k-1}\binom{k-1}{l}$ additional variables to

119   "one-hot encode" for every relevant conditional expectation. For a given obscurable variable, $x_h$,

120   Algorithm 1 adds a variable for every obscuration pattern it could be a part of. We will now show that

121   $\tilde{D}$ could have come from a valid good fisherman setting.

122   **Theorem 3.3** (Reduction to good fisherman self-selection)**.** *Using the same* $\varepsilon_j$ *as those from the*

123   *obscured models, dataset* $\tilde{D}$ *would be the best response to a maximizing self-selection over* $|\mathcal{P}|$ *linear*

124   *models where:* $\tilde{f}_j(\tilde{\mathbf{x}}^{(i)}) := w_0 + \tilde{\mathbf{w}}_j^\top \tilde{\mathbf{x}}^{(i)} + \varepsilon_j$, *each* $\tilde{\mathbf{w}}_j$ *is constructed according to Algorithm 2, and*

$$w_0 := \sum_{h \in [k]} w_h \left(-\mathbb{1}_{w_h \geq 0}|\min\{0, l_h\}| + \mathbb{1}_{w_h < 0}|\max\{0, u_h\}|\right)$$

125   To prove this, we need to show that for every agent of $\tilde{D}$, a best response in this good fisherman setting

126   would indeed still be the $j^\star$th model and the $j^\star$th model would produce that outcome. The intuition of

127   this result can be seen directly from the following lemma statements. First, the transformed features,

128   when multiplied by the $\tilde{\mathbf{w}}_{j^\star}$ and added to $\varepsilon_j + w_0$, produce the same outcome as $f_{j^\star}(\mathbf{x}^{(i)})$!

129   **Lemma 3.1** (Output of $j^\star$ model is stable)**.** *For a point,* $\mathbf{x}_{j^\star}^{(i)}$ *we have:* $\tilde{f}_{j^\star}(\tilde{\mathbf{x}}^{(i)}) = f_{j^\star}(\mathbf{x}_{j^\star}^{(i)})$

130 Second, due to the construction of $\tilde{\mathbf{x}}^{(i)}$ and $\tilde{\mathbf{w}}_{j'}$ for all $j' \neq j^\star$, the inner product corresponding to
131 each good fisherman model $+\varepsilon_{j'} + w_0$, will yield either the same output or less than the private tests
132 the agent did for obscuration pattern $j'$.

**Lemma 3.2** (Output of $j'$ models is lowered). *For a point,* $\mathbf{x}_{j^\star}^{(i)}$: $\tilde{f}_{j'}(\tilde{\mathbf{x}}^{(i)}) \leq f_{j'}(\mathbf{x}_{j'}^{(i)}) \quad \forall j' \neq j^\star$

134 With these lemmas, the proof of Theorem 3.3 is very direct, clearly

$$\tilde{f}_{j^\star}(\tilde{\mathbf{x}}^{(i)}) = f_{j^\star}(\mathbf{x}_{j^\star}^{(i)}) \geq f_{j'}(\mathbf{x}_{j'}^{(i)}) \geq \tilde{f}_{j'}(\tilde{\mathbf{x}}^{(i)}) \quad \forall j' \neq j^\star$$

135 Thus $j^\star$ is the best response and we still have the same $y^{(i)}$!

## 3.2 Estimating w

137 After converting the dataset to one that could be the result of a maximum selection problem over
138 linear models, with a few additional assumptions, the learner can run the algorithm presented by
139 Cherapanamjeri et al. [1] to estimate $\tilde{\mathbf{w}}_j \quad \forall j \in |\mathcal{P}|$ and thus have estimates for $\mathbf{w}$! Recall that from
140 Algorithm 2, we know which elements of $\tilde{\mathbf{w}}_j$ are equivalent to which elements of $\mathbf{w}$, so we can
141 directly construct good estimates of $\mathbf{w}$ from good estimates of $\tilde{\mathbf{w}}_j$.

**Corollary 3.1** (Corollary of Thm 3.3 and Thm 1 [1] ). *Let* $\{\mathbf{x}_{j^\star}, y^{(i)}, j^{\star(i)}\}_{i \in [n]}$ *be $n$ observations*
143 *from an obscurable fisherman model as described in Section 2. Let $\hat{\mathbf{w}}$ be the estimator of the $\mathbf{w}$. Given*
144 *assumptions 3.1 and 3.2, as well as the additional assumptions 1, 2, and 3 from Cherapanamjeri et al.*
145 *[1], there exists an algorithm such that with probability at least .99,*

$$\|\mathbf{w} - \hat{\mathbf{w}}\|_2^2 \leq poly(\sigma, |\mathcal{P}|, 1/\alpha, B, C) \frac{\log n}{n}$$

146 *under $poly(n, g(k, d), |\mathcal{P}|, 1/\alpha, B, C, \sigma, 1/\sigma)$ running time.*

147 *Where $\alpha, B, C$ are constants defined by assumptions 1, 2, and 3 from Cherapanamjeri et al. [1]*

148 Unfortunately, in the parameters of the obscured problem, this is not a very efficient result. Recall
149 that $|\mathcal{P}| = \sum_{l=0}^{k} \binom{k}{l}$ and $g(k, d) := k \sum_{l=0}^{k-1} \binom{k-1}{l} + d$. The number of obscuration patterns, i.e.,
150 models and the number of variables is combinatorial in the number of obscurable variables, which
151 could be as large as $d - 1$!

# 4 Conclusion and Future Work

153 We present a model of agents being able to self-select their set of obscurable features. We provide
154 preliminary results of the estimation of linear model coefficients despite the selection bias that arises
155 from strategic obscuration. Estimation in this setting can be viewed with both a missing not at
156 random (MNAR) problem lens and model self-selection lens. Importantly, under the model-selection
157 perspective, we can reduce the problem to a high-dimensional version of good fisherman setting[1].
158 Unfortunately, the reduction increases the number of data dimensions such that known algorithms will
159 not be efficient in the original dimensions of the problem. Further, the reduction requires knowledge
160 of conditional expectations, which is a strong assumption.

161 In the extended work, we hope to prove an alternate $\mathbf{w}$ estimation method through a more direct MLE
162 estimation similar to that which done by Cherapanamjeri et al. [1]. Because the presented result has
163 shown that the obscurable fisherman setting could be reduced to a version of the good fisherman one,
164 it may be that there exists an analogous population likelihood function that is strongly concave with a
165 stationary point at $\mathbf{w}$, which could be approximately optimized via SGD. Alternatively, as there is a
166 combinatorial (in $d$) variable problem in the reduction, there may be applications of kernelization that
167 remove this issue.

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

# A   Supplementary material

## A.1   Supplementary material for Section 2

### A.1.1   Example

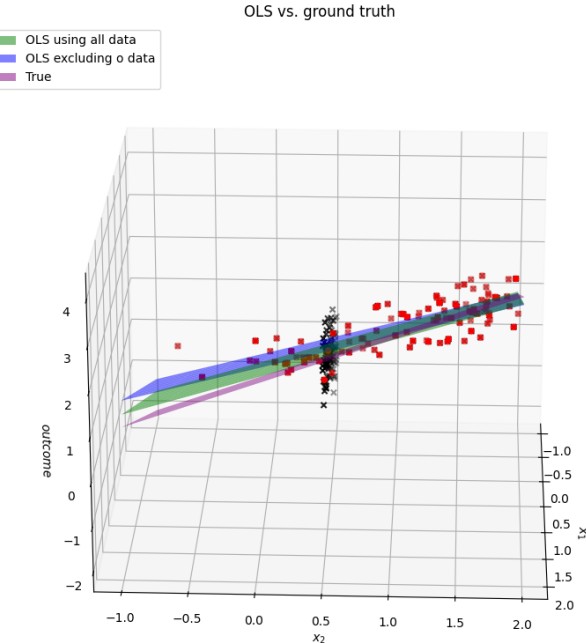

Figure 1: Learner runs OLS on $n = 200$ datapoints detailed in Example 2.1. Black Xs represent the points with obscured $x_2$ elements (missing $x_2$ is imputed as .5 for the green OLS). Red points represent those which are not obscured at all.

 ## A.2 Supplementary material for Section 3

 ### A.2.1 Algorithms to compute the reduction

---

**Algorithm 1** Construct $\tilde{\mathbf{x}}^{(i)}$

---

**Require:** $\mathbf{x}_{j^\star}^{(i)}; \mathbb{E}[x_h | U(\mathcal{O}_j)] \quad \forall h \in \mathcal{O}_j, \forall j \in |\mathcal{P}|; l_h, u_h \quad \forall h \in [k]$

   $\tilde{\mathbf{x}}^{(i)} = []$

   **for** $h \leftarrow 1$ to $k$ **do**                                       ▷ loop adds $k$ elements

      **if** $x_{j^\star,h}^{(i)} \neq o$ **then**

         **if** $w_h \geq 0$ **then**

            **Append** $x_{j^\star,h}^{(i)} + |\min\{0, l_h\}|$ **to** $\tilde{\mathbf{x}}^{(i)}$     ▷ if not obscured, add shifted known value

         **else**

            **Append** $x_{j^\star,h}^{(i)} - |\max\{0, u_h\}|$ **to** $\tilde{\mathbf{x}}^{(i)}$     ▷ if not obscured, add shifted known value

      **else**

         **Append** $0$ **to** $\tilde{\mathbf{x}}^{(i)}$                       ▷ if obscured, add 0

   **for** $h \leftarrow 1$ to $k$ **do**                        ▷ loop adds $k \sum_{j=0}^{k-1} \binom{k-1}{j}$ elements

      **for** $l \leftarrow 0$ to $k-1$ **do**

         **for all** $S \in \binom{[k]\setminus\{h\}}{l}$ **do**        ▷ loop through all obscuration patterns that include $h$

            $\mathcal{O} \leftarrow S \cup [h]$                   ▷ construct obscuration pattern

            $U(\mathcal{O}) \leftarrow \{x_{j^\star,u}^{(i)} | u \in [d] \setminus \mathcal{O}\}$         ▷ construct set of unobscured elements

            **if** $U(\mathcal{O})$ contains elements s.t. $x_{j^\star,u}^{(i)} = o$ **then**      ▷ check if these unobscured are $o$

               **Append** $0$ **to** $\tilde{\mathbf{x}}^{(i)}$          ▷ if yes, then conditional exp incomputable

            **else**

               **if** $w_h \geq 0$ **then**

                  **Append** $\mathbb{E}[x_h | U(\mathcal{O})] + |\min\{0, l_h\}|$ **to** $\tilde{\mathbf{x}}^{(i)}$     ▷ if no, add shifted cond exp

               **else**

                  **Append** $\mathbb{E}[x_h | U(\mathcal{O})] - |\max\{0, u_h\}|$ **to** $\tilde{\mathbf{x}}^{(i)}$     ▷ if no, add shifted cond exp

   **for** $u \leftarrow k+1$ to $d$ **do**                       ▷ loop adds $d - k$ elements

      **Append** $x_u^{(i)}$ **to** $\tilde{\mathbf{x}}^{(i)}$                     ▷ add unobscured value

   **return** $\tilde{\mathbf{x}}^{(i)}$                             ▷ constructed feature vector $\in \mathbb{R}^{g(k,d)}$

---

**Algorithm 2** Construct $\tilde{\mathbf{w}}_j$ to match obscuration pattern, $\mathcal{O}_j$

---

**Require:** $\mathcal{O}_j$, the obscuration pattern of model $j$

   $\tilde{\mathbf{w}}_j = []$

   **for** $h \leftarrow 1$ to $k$ **do**                           ▷ loop adds $k$ elements

      **if** $h \in \mathcal{O}_j$ **then**

         **Append** $0$ **to** $\tilde{\mathbf{w}}_j$         ▷ if $h$ is obscured in this model don't turn on $w$

      **else**

         **Append** $w_h$ **to** $\tilde{\mathbf{w}}_j$             ▷ if $h$ is in this model turn on $w$

   **for** $h \leftarrow 1$ to $k$ **do**                   ▷ loop adds $k \sum_{j=0}^{k-1} \binom{k-1}{j}$ elements

      **for** $l \leftarrow 0$ to $k-1$ **do**

         **for all** $S \in \binom{[k]\setminus\{h\}}{l}$ **do**        ▷ loop through all obscuration patterns that include $h$

            $\mathcal{O} \leftarrow S \cup [h]$

            **if** $\mathcal{O} = \mathcal{O}_j$ **then**

               **Append** $w_h$ **to** $\tilde{\mathbf{w}}_j$       ▷ if this conditional exp is in this model, turn on $w$

            **else**

               **Append** $0$ **to** $\tilde{\mathbf{w}}_j$     ▷ if this conditional exp is not in this model, don't turn on $w$

   **for** $u \leftarrow k+1$ to $d$ **do**                     ▷ loop adds $d - k$ elements

      **Append** $w_u$ **to** $\tilde{\mathbf{w}}_j$        ▷ unobscurable vars are always in the model, so always have their coefficients on.

   **return** $\tilde{\mathbf{w}}_j$

---

 **A.2.2 Missing proofs**

 *Proof of Lemma 3.1.* First, note that Algorithm 1 shifts all obscurable variables, $x_h$ by
 $\mathbb{1}_{w_h \geq 0}|\min\{0, l_h\}| - \mathbb{1}_{w_h < 0}|\max\{0, u_h\}|$ and then $\tilde{f}_j$ adds a constant term

$$w_0 := \sum_{h \in [k]} w_h \left(-\mathbb{1}_{w_h \geq 0}|\min\{0, l_h\}| + \mathbb{1}_{w_h < 0}|\max\{0, u_h\}|\right)$$

We can also do this without changing the outcome of any model (or model selection) to the obscurable
setting because this is equivalent to adding and subtracting terms. For the remainder of the proof, we
will refer to this affine version of the model (with $w_0$) and treat the obscurable variables from $D$ as if
they are shifted.

First we shall consider the function of Algorithm 1 and 2. Notice that, for every $i$, Algorithm 1
constructs a vector such that the first $k$ elements correspond to [shifted] actual values of $x_h$ where
possible. Then $k \sum_{j=0}^{k-1} \binom{k-1}{j}$ elements are added to correspond to every obscurable variable's possi-
ble [shifted] conditional expectation. Finally $d - k$ elements at the end are simply the unobscurable
values that must be present. Algorithm 2 on the other hand follows the same construction pattern,
but instead, for a given $\mathcal{O}_j$, or equivalently, for an given model, places a $w_h$ in the element spot
that represents which conditional expectation (or unobscured value) appears in the model. This is
conceptually very similar to a one-hot encoding!

Thus, for $\mathcal{O}_{j^\star}$, Algorithm 2 constructs a $\tilde{\mathbf{w}}$ that

1. For unobscurable variables, indexed by $u$, assigns $\tilde{w}_u = w_u$ to $\tilde{\mathbf{x}}$ element slots corresponding
   to each said unobscurable variable

2. For each obscurable variable, indexed by $h$, only assigns $\tilde{w}_h = w_h$ to the $\tilde{\mathbf{x}}^{(i)}$ element slot
   corresponding to obscurable variable OR conditional expectation appearing in the given
   $\mathbf{x}_{j^\star}^{(i)}$.

As a result,

$$\varepsilon_{j^\star} + w_o + \tilde{\mathbf{w}}_{j^\star}^\top \tilde{\mathbf{x}}^{(i)} = \varepsilon_{j^\star} + w_o + \mathbf{w}^\top \hat{\mathbf{x}}_{j^\star}^{(i)}$$

Where $\hat{\mathbf{x}}_{j^\star}^{(i)}$ is the *shifted* version of the expected feature vector corresponding to obscured feature
vector. This is equivalent to the statement in the lemma. $\square$

*Proof of Lemma 3.2.* As in the proof of Lemma 3.1, note that Algorithm 1 shifts all obscurable
variables, $x_h$ by $\mathbb{1}_{w_h \geq 0}|\min\{0, l_h\}| - \mathbb{1}_{w_h < 0}|\max\{0, u_h\}|$ and then $\tilde{f}_j$ adds a constant term

$$w_0 := \sum_{h \in [k]} w_h \left(-\mathbb{1}_{w_h \geq 0}|\min\{0, l_h\}| + \mathbb{1}_{w_h < 0}|\max\{0, u_h\}|\right)$$

We can also do this without changing the outcome of any model (or model selection) to the obscurable
setting without changing the outcome of any model (or model selection) because this is equivalent to
adding and subtracting terms. For the remainder of the proof, we will refer to this affine version of
the model (with $w_0$) and treat the obscurable variables from $D$ as if they are shifted.

First we shall consider the function of Algorithm 1 and 2. Notice that, for every $i$, Algorithm 1
constructs a vector such that the first $k$ elements correspond to [shifted] actual values of $x_h$ where
possible. Then $k \sum_{j=0}^{k-1} \binom{k-1}{j}$ elements are added to correspond to every obscurable variable's possi-
ble [shifted] conditional expectation. Finally $d - k$ elements at the end are simply the unobscurable
values that must be present. Algorithm 2 on the other hand follows the same construction pattern,
but instead, for a given $\mathcal{O}_j$, or equivalently, for an given model, places a $w_h$ in the element spot
that represents which conditional expectation (or unobscured value) appears in the model. This is
conceptually very similar to a one-hot encoding!

An important nuance happens when the obscuration pattern of $\tilde{\mathbf{w}}_{j'}$ does not match the obscuration
pattern implicit to $\mathbf{x}_{j^\star}^{(i)}$. Algorithm 1 sets as 0 any elements of $\tilde{\mathbf{x}}^{(i)}$ that represent conditional
expectations (or obscurable values) that cannot be computed from $\mathbf{x}_{j^\star}^{(i)}$, which may have missing
values. For example, if $\mathbf{x}_{j^\star}^{(i)} = (o, 1, 4)$ and the first two variables obscurable, one of the elements in

the corresponding $\tilde{\mathbf{x}}^{(i)}$ will be for $\mathbb{E}[x_2|x_1 =?, x_3 = 4]$, but this will be incomputable since obviously $x_1$ is obscured.

Consider an arbitrary element $\tilde{x}_q^{(i)}$ associated with the obscurable element at index $h$. That is, element at index $q$ of $\tilde{\mathbf{x}}$ is some conditional expectation or value of obscurable element at index $h$ of $\mathbf{x}^{(i)}$. As a result of Algorithm 1, if this conditional expectation or value is incomputable as a result of the the obscuration pattern of $\mathbf{x}_{j^\star}^{(i)}$ because relevant values are missing, $\tilde{x}_q^{(i)} = 0$. This means, for any obscuration patterns, $\mathcal{O}_{j'}$, that $\tilde{x}_q^{(i)}$ is represented in, while Algorithm 2 will construct a $\tilde{\mathbf{w}}$ that sets $\tilde{w}_q = w_h$, $\tilde{w}_q \tilde{x}_q^{(i)} = 0$! Meanwhile, in the earlier private test for that obscuration done by the agent, she tested $\varepsilon_{j'} + w_o + \mathbf{w}^\top \hat{\mathbf{x}}_{j'}^{(i)}$, and she would have:

$$w_h \hat{x}_{j',h} = w_h \left( \mathbb{E}[x_h|U(\mathcal{O}_{j'})] + \mathbb{1}_{w_h \geq 0}|\min\{0, l_h\}| - \mathbb{1}_{w_h < 0}|\max\{0, u_h\}| \right) \geq 0$$

(again, for this proof we redefine $\hat{\mathbf{x}}_{j'}$ as the *shifted* expected feature vector) because she had access to missing variables and by construction of the shift its greater than or equal to zero. As a result we see that:

$$\varepsilon_{j'} + w_o + \mathbf{w}^\top \tilde{\mathbf{x}}_{j'}^{(i)} \leq \varepsilon_{j'} + w_o + \mathbf{w}^\top \hat{\mathbf{x}}_{j'}^{(i)} \quad \forall j' \neq j^\star$$

Where $\hat{\mathbf{x}}_{j^\star}^{(i)}$ is the *shifted* version of the expected feature vector corresponding to obscured feature vector. This is equivalent to the statement in the lemma. $\qquad \square$

*Proof of Theorem 3.3.* We need to show that, for every $i$, were $\tilde{\mathbf{x}}^{(i)}$ the underlying true features generated, then $\max_{j \in |\mathcal{P}|} \tilde{f}_j(\tilde{\mathbf{x}}^{(i)})$ would generate the $y^{(i)}$ and the $j^{\star(i)}$ given. Equivalently, that a best response would indeed be the $j^\star$th model and the $j^\star$th model would produce that outcome.

The result directly follows from Lemma 3.1 and 3.2. First, Lemma 3.1 confirms that for all agents $i$, model $j^\star$ does yield the same output under both $f_{j^\star}$ and $\tilde{f}_{j^\star}$ settings. All that remains to show is that $\tilde{f}_{j^\star}$ is in fact the best outcome of all $\tilde{f}_j$. Notice that for an point $\mathbf{x}_{j^\star}^{(i)}$, we know that $f_{j^\star}(\mathbf{x}_{j^\star}^{(i)}) > f_{j'}(\mathbf{x}_{j'}^{(i)}) \quad \forall j' \neq j^\star$ because the agent selected $j^\star$. From Lemmas 3.1 and 3.2:

$$\tilde{f}_{j^\star}(\tilde{\mathbf{x}}^{(i)}) = f_{j^\star}(\mathbf{x}_{j^\star}^{(i)}) > f_{j'}(\mathbf{x}_{j'}^{(i)}) \geq \tilde{f}_{j'}(\tilde{\mathbf{x}}^{(i)}) \quad \forall j' \neq j^\star$$

Thus $j^\star$ is the best response in the transformed good fisherman model set as well! $\qquad \square$

