# OpenReview forum: "Obscurable Fishermen"
_NeurIPS.cc/2025/Workshop/Reliable_ML — NeurIPS 2025 - Reliable ML Workshop_

### Official Review · Reviewer_MoCZ · 2025-09-09
**Preliminary work, writing in Section 3 needs to improve significantly**

**Rating:** 6
**Confidence:** 4

**Review:**

The paper tries to address a problem of "adverse selection" in the sense that there is some missing elements from "features", but the missing elements are not randomly distributed. The features determine the outcome from a class of linear models, with a separate model for each possible missing data pattern. The missing data is chosen to maximize the output among the various linear models. The paper tries to estimate the linear model parameters from the data.

*Strengths*: It is an interesting problem, and the authors have made some progress.

*Weaknesses*: (1) Please clarify the distinctions in algorithms/theorems from Cherapanamjeri et al. [1]. The paper relies on it heavily so it would help to discuss their results and techniques.

(2) Example 2.1 is not a sufficient justification to prove OLS is biased, i.e., doing $1$ trial over $n$ samples and showing deviation from true value does not prove bias of an estimator.  Please provide theoretical justification, it should not be too difficult for the example considered.

(3) Section 3 can be written in a better manner. The algorithms and the results could be described better for clarity.

---

### Official Review · Reviewer_ggT5 · 2025-09-13
**Review of NeurIPS 2025 Workshop Reliable ML Submission152 - "Obscurable Fishermen"**

**Rating:** 9
**Confidence:** 4

**Review:**

# Summary
The author(s) introduce the "obscurable fishermen" problem, where the learner collects a set of linear regression data $(x^{(i)}, y^{(i)}), i\in [n]$ from a set of agents with the goal of estimating the (single) unknown linear regressor.
The twist is that each agent can strategically omit a subset of the first $k$ out of $d$ features such that the reported linear regression output $y^{(i)}$, computed with imputed missing features based on the conditional experience given observed features, will yield the largest $y^{(i)}$.

The author(s) then present a clever reduction to the "known-index linear regression with self-selection bias" problem (a.k.a., "good fisherman" problem) defined by Cherapanamjeri et al. [1].
In this variant of linear regression, there are multiple unknown regressors $w_1, \dots, w_k$ and each agent with feature $x^{(i)}$ observes the potential outcomes $y_j^{(i)} = \langle x^{(i)}, w_j \rangle + \varepsilon_j$ for $\varepsilon_j\sim \mathcal{N}(0, \sigma^2)$.
However, the agent only reports the maximum output $y_{max}^{(i)} = \max_j y_j^{(i)}$ and the maximizing index $\bar j^{(i)} = argmax_j y_j^{(i)}$, but the learner is still expected to estimate all the unknown regressors using only the self-selected data $(x^{(i)}, y_{max}^{(i)}, \bar j^{(i)}), i\in [n]$.


## Reduction
The reduction as presented is somewhat difficult to parse, but, is easily illustrated with an example.
Suppose
- the features are GPA, SAT score, and years of experience (Exp)
- agent $i$ has features $(3.8, 1300, 1)$ ($d=3$) and can omit any subset of the first $k=2$ features.
- the unknown linear regressor is $w = (10, 0.02, 5000)$
- the agent strategically reports $(3.8, o, 1)$, where $o$ indicates an omitted feature.

Then, the learner proceeds as follows:
1. The learner constructs a large feature vector $\tilde x^{(i)}$ of all possible features and imputed features.
In other words, $\tilde x^{(i)}$ corresponds to (GPA, SAT, GPA|SAT+Exp, GPA|Exp, SAT|GPA+Exp, SAT|Exp, Exp), where, for example, GPA|SAT+Exp indicates the expected GPA given the SAT and Exp features.
1. For each possible way of omitting features (indexed by $j$), we can pretend this corresponds to a sparse linear regressor $\tilde w_j$ constructed as follows.
    1. $\tilde w_0 = (w_{GPA}, w_{SAT}, 0, 0, 0, 0, w_{Exp})$ corresponding to no omission
    1. $\tilde w_1 = (0, w_{SAT}, w_{GPA}, 0, 0, 0, w_{Exp})$ corresponding to omission of GPA
    1. $\tilde w_2 = (w_{GPA}, 0, 0, 0, w_{SAT}, 0, w_{Exp})$ corresponding to omission of SAT
    1. $\tilde w_3 = (0, 0, 0, w_{GPA}, 0, w_{SAT}, w_{Exp})$ corresponding to omission of both GPA, SAT
Note that the learner does not need to actually construct all $\tilde w_j$'s but can estimate the true underlying $w$ given estimates of $\tilde w_j$'s.
1. Run the algorithm of Cherapanamjeri et al. [1] pretending the input was $(\tilde x^{(i)}, y^{(i)}, j_i)$ where $j_i$ is the index of the omission pattern.

## Caveats
- In order to construct $\tilde x^{(i)}$, we must assume access to a conditional expectation oracle for features.
- Moreover, if the agent omits some features, it is not possible to call the conditional expectation for all omission patterns.
However, this can be resolved by appropriately shifting constructed features and adding a bias term.

# Positives
- This seems like a novel and practically motivated problem with an interesting theoretical model.
- The algorithmic idea to reduce to "good fishermen", while inefficient, is clever and not straightforward, and seems like a nice workshop contribution. I especially like the clever "shifting" workaround for computing $\tilde x^{(i)}$ when some components cannot be imputed due to missing data.
- There is a clear direction for future work to overcome the computational inefficiency.
If this were possible, it would significantly strengthen the contribution.
The technical presentation is pretty meticulous, and the amount of detail is appropriate.

# Comments & Concerns
1. While the details of the reduction algorithm is clearly presented, the high-level intuition is severely lacking and the pseudocode is very hard to parse for this reason. Adding an example as I illustrated above would make the reading much smoother.
1. It seems the learner must also know the sign $\text{sign}(w_h)$ of the linear regressor coefficient corresponding to the $h$-th possible missing features in order to compute the right shift and bias term in the expanded vector $\tilde x^{(i)}$.
1. Even if the original features $x^{(i)}$ satisfy the standard fixed-design linear regression assumptions from Cherapanamjeri et al. [1], it is not clear the expanded dataset $\tilde x^{(i)}$ satisfy those conditions, especially the thickness condition $\sum_i \tilde x^{(i)} [\tilde x^{(i)}]^\top \succeq I$.
1. Related to my point above and the inefficiency issue, it might be interesting to see if the sparsity of the hypothetical unknown regressors $\tilde w_j$ can be exploited to design faster algorithms.
1. Some other works on self-selection [GM24;KMZ25] may be of interest. Specifically, [KMZ25] presents a general likelihood optimization framework that may directly capture the obscurable fisherman problem without needing to perform the expensive reduction to good fishermen.

# Nitpicks
- Line 79: $|\mathcal{P}|$ should be $\mathcal P$
- Line 114: The definition is confusing without quantifiers on $u_h, l_h$. I think it should be a given $u_h, l_h$ for all features
- Line 115: I think $\tilde D$ should be defined as $\tilde D = \lbrace (1, \tilde x^{(i)}), y^{(i)}, j^{*(i)} \rbrace$

---
# References
[GM24] Jason Gaitonde, and Elchanan Mossel. "Sample-efficient linear regression with self-selection bias." arXiv preprint arXiv:2402.14229 (2024)

[KMZ25] Alkis Kalavasis, Anay Mehrotra, and Felix Zhou. "Can SGD Select Good Fishermen? Local Convergence under Self-Selection Biases and Beyond." arXiv preprint arXiv:2504.07133 (2025).

---

### Official Review · Reviewer_pHwE · 2025-09-17
**An interesting question on statistical estimation from strategic self-selection**

**Rating:** 7
**Confidence:** 3

**Review:**

Summary: This paper asks an interesting question inspired by the "what makes a good fisherman" problem. In this model, the strategic agents can choose an obscure pattern to hide some coordinates of their feature vector. A fixed linear model produces outcomes by using conditional expectations for these missing coordinates. Then the agent chooses the best obscure pattern that gives the highest outcome. The question is to efficiently estimate the linear model given an obscured dataset containing self-selected obscured features and outcomes. The authors show that the standard estimation approach fails, but give a reduction to the standard fishermen setting. However, this reduction does not give a polynomial-time algorithm since the number of variables increases exponentially.

Comment: I think the problem of obscure fishermen is interesting and relevant to the topic of the workshop. The weakness of the paper is that it only gives an inefficient reduction to the standard fisherman setting.